# Women's Perceptions of Nature: An Ecofeminist Analysis of Tsitsi Dangarembga's *This Mournable Body*

Nigus Michael Gebreyohannes * and Abiye Daniel Ambachew *

Department of Foreign Languages and Literature, Addis Ababa University, Addis Ababa P.O. Box 1176, Ethiopia
* Correspondence: nigus.michael@gmail.com (N.M.G.); abiye_daniel@yahoo.com (A.D.A.)

**Abstract:** The purpose of this article is to explore ecofeminist issues in Tsitsi Dangarembga's *This Mournable Body*. It mainly focuses on the relationship between women and nature and explores the perceptions of women toward the natural environment. Thus, a close reading was done to extract the necessary information from the novel. Next, the extracted data was analyzed using textual analysis. Additionally, ecofeminist literary criticism was adopted as a lens to analyze the novel. Therefore, based on the analysis made, the novel portrays various issues related to women and nature. Firstly, the novel shows that African women are gardeners, agricultural laborers, and protectors of the land and the natural environment, which makes them have a strong relationship with the natural environment. On the other hand, it shows, that not all women have the same perception of nature. In this manner, Tracey, a white businesswoman, considered nature as an income generator in the form of the ecotourism industry, regardless of the degradation of the natural environment. In contrast, the native women consider nature as a means of their survival. Nyasha, a woman from Zimbabwe, believes nature and land space enhance co-operation and harmony between inhabitants. Similarly, Tambudzai, also from Zimbabwe, recounts the beauty and healing power of nature, and she expresses her concern about the degradation of the natural environment. Therefore, the novel has discovered the different relationships between women and nature. Their understanding of and connection to nature vary and directly relate to their background and context. At last, the novel portrays the impact of neocolonialism and capitalism predominantly on women and nature. In this manner, the author shows her concern for African women and the natural environment.

**Keywords:** women; nature; perception; ecofeminism; neo-colonialism; degradation; gardening

## 1. Introduction

Ecofeminism is becoming an important field in both contemporary feminist and environmental studies. As the literature show, it is originated from various backgrounds but predominantly from the global feminism movement and environmental activism. Since Francoise d'Eaubonne in 1974 coined the name, it has found diverse expressions in the arts, literature and language, science and technology, philosophy and religion, and nongovernmental organizations (NGOs).

Ecofeminism as an academic body of work peaked in the early 1990s, when philosophers, such as Karen Warren and Val Plumwood, released volumes that described the theoretical structures underlying ecofeminist philosophy (Abatemarco 2018). Thus, Plumwood (1993) posits ecofeminism more focuses on four types of exploitation: race, class, gender, and nature. Warren (1997a) focuses on important connections between how one treats women, people of colour, and the underclass on one hand and how one treats the nonhuman natural environment on the other. In this book, Warren provides the examination of ecofeminism from a variety of cross-cultural and multidisciplinary perspectives.

Scholars such as Vandana Shiva have contributed such a vital role in positioning the theory from the South. Shiva's *Staying Alive* (Shiva 1988) is the first grounding ecofeminist book that links ecological crises, colonialism, and the oppression of women. Moreover, Mies

and Shiva (1993) also wrote a book on ecofeminism that argues for accepting boundaries, rejecting the commoditization of needs, and committing to a new ethics. They question prevalent economic theories, traditional notions of women's emancipation, the myth of 'catching up' development, the philosophical foundations of modern science and technology, and the omission of ethics when discussing so many issues, including advances in reproductive technology and biotechnology. Moreover, to many of its advocates, ecofeminism is seen as an innovative response to contemporary relations of domination, a force with which to negotiate a new era of gendered and natured relations (Sandilands 1999).

Ruether (2004) also goes on to explain how corporate globalization, as the dominant economic power, is one challenge that is exacerbating environmental destruction, undermining authentic democracy, undermining cultural diversity, undermining social integrity, and widening the gap between rich and poor around the world. She claims that world religions have understood the potential environmental harm caused by their traditions, and that gender inequalities in religion and culture have been linked to human–nature hierarchies. As a result, they have begun to envisage a novel way of understanding human nature as an interrelated and life-giving matrix. In this approach, she confirms the relationship between feminism, ecology, and religion theological philosophy as having considerable relevance to ending women's dominance over the environment and several other issues.

Reuther's and other spiritual ecofeminists' (such as Starhawk 2004) views of nature can be essentially compared to that of precolonial Africans, who believed in the spirituality of the earth and nature in general. This is still practiced by some African tribes to this day. This shows that African ecofeminism is not just a theory or a worldview. It is rather existing as a way of life that has been practiced for many years by indigenous people, especially by women. The reason is that, in many African cultures, women are the protectors and guardians of the earth, similar to goddesses. This version of relationship is unique because it is formed by indigenous African cultures and beliefs. For example, the Gikuyu people in Kenya see women as divine beings who are responsible for maintaining and protecting the earth (Kenyatta 1965). Furthermore, the women of the Akan tribe in Ghana, who are responsible for bearing children, raising them, and taking care of the community, are often committed to the Earth and its capacity to provide for the needs of the tribe's survival (Molato 2020). The Shona people in Zimbabwe are also known for their connectedness to the natural environment. Women in Shona society had a close relationship with the environment because they were the ones who gathered fruit and firewood and collected water from the water sources. It is women who, on a daily basis, were confronted with the need to observe taboos associated with the environment (Manyonganise and Museka 2020).

However, during the colonial period, women and the environment were more victims of colonizers and their administrative systems. Women's power to control the land and their connection to it was distorted. The colonial authorities preferred to build coalitions with African men in order to increase patriarchal control over women. As a result, as a colonial-capitalist economy was forcibly imposed, the political and economic standing of women changed dramatically at the same time that expropriation of land occurred, and socio-political rights were increasingly denied or eliminated. Capitalism demanded that the reserves (land designated for black settlement and cultivation) serve as a labor pool, supplying the industrializing cities and, to the greatest extent feasible, feeding their own populations (Daymond et al. 2003).

Nevertheless, with many of its hardships, African women in the post-colonial era have been struggling to restore their historic status in their society and their connection to the natural environment. They have been confronting the degradation of the natural environment and the climate change caused by human and natural phenomena. In this case, Wangari Mathai of Kenya is the most known environmental activist, having inspired millions of women to plant trees across Kenya. She established a movement called the Green Belt Movement. As Maathai (2003, p. 6) explains, "the Green Belt Movement is a grassroots non-governmental organization (NGO) that focuses on environmental conservation and

development". It does this mainly through a nationwide grassroots tree-planting campaign that is its core activity.

Along with campaigners, such as Wangari Maathai, some African writers are beginning to depict the environmental catastrophe and its effects on humans and non-human living things. Esamagu (2020, p. 243) added that "in recent times, there is a surge of African writings chiefly committed to preserving the earth . . . " Based on this evidence, environmental concern is becoming one of the themes addressed in postcolonial literature. Thomas (2009, p. 230) describes the major themes of contemporary African literature as: "concern for the environment, immigration, democratization, civil conflict, genocide, the National Conferences, child soldiers, the disintegration of the nation state, AIDS, the Truth and Reconciliation Commission, and globalization". In addition to this, African women's writing has brought to light a variety of gender issues in African society. Their works predominantly focuses on social conditions related to problems, such as polygamy and male dominance, as well as their search for happiness and a fulfilling place in contemporary society through access to education and full participation in the economic and national politics of the new African states (Irele 2009).

However, in African literature, critics have overlooked gender as an asocial and analytic category (Stratton 2020). Furthermore, because African literary criticism and research concentrate solely on human agents, the non-human natural environment is relatively neglected, with the exception of the current growing research being published, (Iheka 2018). Therefore, this shows that there is still a gap in exploring the portrayal of women and the natural environment in African literature. This implies that, so far, little attention has been paid to ecofeminist issues in African literature in general.

Thus, this article examines ecofeminist issues in Tsitsi Dangarembga's *This Mournable Body* (Dangarembga 2018). Tsitsi Dangarembga is among the most renowned female writers in Africa. Her novel, *Nervous Conditions* (Dangarembga 1988), is her most influential, widely read, and studied novel, and *This Mournable Body* (Dangarembga 2018) is another novel that is a continuation of Tambudzai's stories from the two previous novels: *Nervous Condition* (Dangarembga 1988) and *The Book of Not* (Dangarembga 2006). *This Mournable Body* addresses various issues, mostly related to the main character Tambudzai in post-colonial Zimbabwe. However, this research examines the portrayal of the relationship between women, nature, and other indigenous people. Furthermore, it investigates the way women and some other indigenous people perceive the natural environment

Therefore, in order to address these issues, ecofeminist literary criticism, which (Vakoch 2012) calls "Feminist Ecocriticism", has been adopted as tool of analysis. Gaard and Murphy (1998), Carr (2000), Campbell (2009), Vakoch (2012), and Gaard (2010) have defined the notion and scope of ecofeminist literary criticism under various contexts. In addition, Chae (2015) used the term "postcolonial ecofeminism", and argued that it should not only locate a woman–nature connection and society's treatment of both but also critically investigate the women–nature relationship specific to postcolonial nations due to postcolonial women's double bind. This link is influenced by factors such as class, race, religion, location, and politics. Therefore, this research explores the novel based on these premises and considering the various issues of ecofeminism.

In order to obtain relevant data, the novel was taken as a primary source. In addition, many secondary sources, such as reference books and e-journals, were accessed to obtain further information on the topic. In order to extract the data, a close reading of the novel was conducted. Next, textual analysis of extracts was undertaken carefully. Finally, the extracts were analyzed based on the perspective of ecofeminism literary theory. Therefore, the novel portrays the following issues.

## 2. Results and Discussions

### 2.1. The Relationship between Women and Nature

Tsitsi Dangarembga's *This Mournable Body* portrays women and the natural environment in many ways. Women are represented as gardening gurus. Shiva (1988, p. 73)

states that "the backyard of each rural home was a nursery, and each peasant woman is the sylviculturalist" to shows the women's knowledge and skill at gardening. Parallel to this, a widow-woman in the novel is known for growing the finest fresh food in all of Zimbabwe. She is popular because she has the finest garden all of Zimbabwe. The narrator says: "Down in Honey Valley where the finest fresh foods grow. That is one of mine. The whole of Zimbabwe knows it" (Dangarembga 2018, p. 26). In addition, *Tsitsi Dangarembga* demonstrates that a woman has better skills at cultivating gardens, referring to Christine, the niece of a widowed woman, Mai Manyanga, who owns a garden. Tambudzai described her as a hard-working and gardening expert. This signifies women's knowledge of gardening. On the other hand, it clearly reflects the interaction between women and nature. In short, Tambudzai describes as follows how Christine was treating the plants in the garden:

> She emerges testing a couple of catapults. You watch the expertise of her fingers enviously as she strips off lengths of black rubber, repairing the hosepipe from her bounty. Deftly, she lays the hose spout at the highest point of a seedling bed while she swings a hoe. She waters the sweet peas around the widow's cottage. She sweeps the students' slab. She replants a patch of grass under the guava tree. She is a woman who is good at what she does. And this is intriguing. (Dangarembga 2018, p. 37)

Furthermore, Dangarembga reveals that women who live in rural areas have good knowledge about cultivating and taking care of their lands. It shows that women can change their lives by introducing new knowledge about land usage and farming. Mrs. Samhungu is an exemplary woman who changed her life by introducing "new ideas, new ways, new mixtures, and new crops to the pale village soil until it gave up withholding and her garden thrived, and on account of her prowess, which everyone hoped to share, she was voted chairwoman" (Dangarembga 2018, p. 251). This woman was represented as a self-sufficient woman in her own way, which could be a lesson to many women in her community. On the other hand, it shows that women in rural areas of Africa still have a close relationship with nature, just as their foremothers did during the precolonial era.

Moreover, the novel demonstrates how women value organic products. The chairwoman states that she always consumes organic food in the village. She appreciates the benefits of the organic products that she obtains from her land. On the other hand, she warns about the dangers of using fertilizer. She believes that food grown with fertilizer does not have a pleasant taste; it is rather the organic food that is delicious. She also claims that using fertilizer pollutes and destroys the natural environment. As a result, she states that the town's food is neither organic nor tasty. This implies the way she depreciates inorganic food products and values the natural product because it is healthy and tasty. The woman states:

> There is nothing better than tea with milk and sugar and a dish of sweet potatoes, she beams. And these are the very best. We don't put any fertilizer on these ones like they do over there in the town, so they are very good. Eat, Tambudzai", she urges. "You will enjoy them . . . They are delicious, the most delicious Mai agrees. Like sugar and butter. (Dangarembga 2018, p. 249)

In this manner, the author warns against the use of chemical fertilizer, which degrades the land and food products. It demonstrates her concern about the contamination generated by the so-called "modern agricultural mechanism", which promotes the use of chemical fertilizers, herbicides, pesticides, and other agricultural chemicals. She, on the other hand, encourages people to produce and consume organic products since they are nutritious and tasty, and they have no negative impact on the environment.

*This Mournable Body* also shows the myriad responsibilities related to agricultural activities. It is obvious that, as mothers, they are the ones who take care of their families at home. While this is a huge burden, women still take full responsibility for working in agricultural activities. Working in agriculture is the hardest job. However, women

tirelessly spend their time producing crops. They are also the ones engaged in trade and marketing. For example, Tambudzai states that "women who seem stunned by the fact of their existence trudge along the verges, babies on their backs, bags of seed and fertilizer on their heads, or else they simply stand, waiting for combis" (Dangarembga 2018, p. 170). In this way, the author portrays women as being directly connected to the natural world because they are in charge of agricultural duties. They are in charge of caring for and cultivating the land. This demonstrates also the importance of land for women in an African context. This implies that the deterioration of the environment, on the other hand, has a direct impact on them and their families.

*2.2. Women's Perception of Nature*

Tsitsi Dangarembga has also portrayed the interaction between women and the natural environment from different angles. One of the issues in the novel is an ecotourism venture owned by a white businesswoman named Tracy. Her participation in this venture can be viewed from different points of view. For one thing, it shows the positive outlook Tracy has towards the natural environment. This is because participating in the ecotourism industry is not the same as other businesses, which distort and pollute the natural environment, but rather has a notion of giving value to the natural environment with minimum impact. That is the reason Tracey herself states that, "Green Jacaranda beyond that, it is a start-up dealing with environmentally friendly entrepreneurship solutions over a range of programmes" (Dangarembga 2018, p. 170).

The extract also shows that Tracy's setting up of an environmentally friendly venture is a significant business in protecting the natural environment. Additionally, this shows her positive perception of the natural environment while there is still a minimum disturbance of the biosphere by human activities.

In *This Mournable Body*, female characters have been portrayed as good gardeners, which shows how women interact with the natural environment. Women, such as Mia Manyannga, are represented as gardeners who cultivate their gardens with pleasure and hope, which shows their positive attitude toward nature. Not only in this novel, in her previous novel, *The Book of Not*, Tsitsi Dangarembga portrayed gardens: Maiguru's garden, the Harare gardens, the Sacred Heart gardens, and the gardens to show interconnectivity between black women in the narrative, the racial colonial setting, and nature" (Pasi 2016). As a continual of this, in *This Mournable Body*, gardening is represented as a woman's work and a daily practice that is also common in many African societies. On the other hand, it reflects that women are the ones who are closest to the natural environment.

Similarly, *This Mournable Body* demonstrates the significance of space-land. Tambudzai's cousin, Nyasha, believes that space-land fosters collaboration. She believes that there are many things that can be done in space. For one reason, that space can bring about and promote co-operation. "She says you can do things with space", Leon says. "That here she has enough room to make a difference. Her philosophy is that space promotes co-operation, lack of it, fighting" (Dangarembga 2018, p. 129). As she states, space-land gives people freedom and peace. They can cooperate and integrate well when they get enough space.

Tsitsi Dangarembga portrays the sensitivity of indigenous people to protecting their natural environment. Indigenous peoples in general and women in particular have a close connection to their natural environment. Thus, they are ready to fight and protest against the destruction of the ecosystem, for it is the only means for their survival. As Merchant (1995, p. 155) recounts, in 1980, Native American people organized by women community leaders claimed that they must treat nature with love because it provides them with physical sustenance and spiritual power. They note, "We are people of the land. We believe that the land is not to be owned but to be shared. We believe that we are guardians of the land" Merchant (1995, p. 155). Thus, having this concept in mind, Tracy and her friends intend to motivate and organize the indigenous people to fight against climate change. She explains, "This way we can sensitize people to and advocate against climate

change at the same time as we're doing business . . . " (Dangarembga 2018, p. 183). Thus, as stated earlier, even if Tracy's intention is to maximize her profit through ecotourism ventures, she still believes that climate change is a sensitive issue that enables people to mobilize against it.

*This Mournable Body* depicts Tracy as a businesswoman who perceives nature as an income generator. Tracey believes that nature means the weather and the sun. So, if there is weather and sun shine, which she chose for her ecotourism venture, that is nature for her. This is the reason that Tracey believes a conducive environment is one that can be an impetus for running a business. So, she says, "Nature, which naturally also means weather. Sun. As time goes on and things get worse or better, we'll strategize for the next phase" (Dangarembga 2018, p. 188). However, in developing countries, nature means a lot to women. J. Warren (1997b, pp. 5, 6) points out that "in developing countries, women are more dependent than men on tree and forest products. Trees provide five essential elements in these household economies: food, fuel, fodder, products for the home (including building materials, household utensils, gardens, dyes, medicines), and income".

Tracy also recounts the greatness of the weather in Zimbabwe and how climate tourism could be a profitable business. In this way, she commercializes nature. She perceives nature with regard to its value as a climate tourist. Therefore, for her, weather or the climate in general is depicted as a source of income. Therefore, since the environment is fresh and unpolluted, she dreams of controlling the whole country and then going to southern Africa in a tourism venture. In short, she explains her plan as follows.

> Zimbabwe's always going to be here. People are always going to want great weather. In principle, it's our one definitely sustainable resource. Climate tourism is the next big thing. There'll be dozens of ventures like ours in five years' time, but we'll have this country, and if things go the way I intend, even all of southern Africa, covered. (Dangarembga 2018, p. 184)

While Tracey considers nature as a source of income and dreams of controlling all of South Africa, Tambudzai describes the exquisiteness and relieving power of mother nature. She particularly exemplifies Zimbabwe's spectacular terrain and sky, which stand for beauty, relief, and freedom. She reveals the meshed ecosystem and her interconnectedness within it. She says "there is exquisite delight in the ripple of pale gold grass over the plain" (Dangarembga 2018, p. 216) to symbolize nature as a source of enjoyment. Moreover, she confirms that nature symbolizes the endless peace, saying "Immeasurable peace abides in a giraffe's neck curving brown and deep gold against the sky that shines too blue to look at, as in the animal's velvet plucking of foliage" (Dangarembga 2018, p. 218).

Additionally, Tambudzai justifies that she is part of all the natural beauty. She, as a human, is one member of the earth's community and feels delight for that. Her sense of belongingness makes her excited. Thus, she states, "The rumble of a lion's purr, the arc of the tusk of an elephant bull, the calculated flick of a predator bird's wing rekindle awe at the fact that you are part of such an existence" (Dangarembga 2018, p. 216). Here, she senses that she is part of it, part of the natural scene she is enjoying.

Moreover, the novel shows the abundance of natural resources and humans' dependency on them. In another word, the novel shows how nature provides daily consumption for human beings. For example, Tambudzai explains that the tourists eat fresh fish and drink marula liqueur, which is made from the marula tree, and other drinks that are indigenous to the tourist sites. She states:

> In the evenings, long drinks anticipate freshly caught fish grilled on an open fire by the chef with a marinade of mazhanje juice or marula liqueur. There are madora and matemba, and sorghum beer for those who dare later in the night, when dancers entertain the guests in the establishment's central clearing. (Dangarembga 2018, p. 216)

On top of that, the novel portrays that nature is the symbol of freshness and impressiveness. Tambudzai explains that "In the cold season, when even the sun is white, grass

stalks sway gently in the breeze and your visitors catch their breath" (Dangarembga 2018, p. 216). In this situation, Tambudzai introduces the existence of human beings in the safari lands and how they add beauty to the natural environment, but how they put nature in danger. She states that after she smelled the smoke and saw a rim of red glow in the distance on nights, the sky started to smell like home. As Tambudzai recounts the situation:

> The smell of smoke from the estate hands' cooking lingers over the safari lands, and, on some evenings, a rim of red glows in the distance like a full moon rising, over in the villages where people reside, their untended fires crackling and smoking with destruction. On those nights, the sky smells like home. (Dangarembga 2018, p. 216)

Besides, as a tourist site supervisor, Tambudzai describes the natural environment, the animals as a source of beauty. The view she describes through the eye of the camera is so impressive and a magnificent part of nature. She describes the shores of seas, oceans, the purple canopy of jacaranda trees, beneath the shade of acacias in the vastness of the Savannah, revealing the way that Tambudzai understands nature. In this way, she reveals how beautiful the natural environment of Africa is. She expressed her delight at being a part of the tour.

On the other hand, the novel explains that nature is God's habitat. The indigenous people call their land "God's Own Country" because it is not penetrated with human traces except by those who originated there. In this case, the next extract signifies that a place inhabited by diversified animals is God's place. The place where few human beings live and many animals exist shows that there is God there. God lives in nature. Thus, the people believe that their place is God's own country, where diversified creatures can live together. It is the settlement of these wild lives, including native people, that causes them some untended fire destruction. Thus, Tambudzai explains the following:

> Your brochure reads, and you advise your group, that the settlers, in awe, named the sprawling veld God's Own Country. The clients exclaim and question each other over every animal track encountered, dung beetle chuckled over, and pool that looked refreshing but might harbour bilharzia. (Dangarembga 2018, p. 216)

Moreover, the natural environment is represented as a source of cheerfulness and national pride. Tambudzai feels happy and proud thinking about the biodiversity of Zimbabwe. Zimbabwe is one of the richest and most beautiful countries in Africa, endowed with a natural environment. It is home to numerous tree species and plenty of wildlife. The country has a number of national parks with various safari animals, such as elephants, lions, and hippos. Moreover, the most internationally known, Victoria Falls and the Zambezi River, are found in Zimbabwe. It is also rich in minerals such as diamonds, gold, coal, iron ore, chromium ore, vanadium, asbestos, nickel, copper, lithium, tin, and platinum group metals. All of these combine to make Zimbabwe the most beautiful and resource-rich country in Africa. However, the country is still poor for different reasons. Thus, the greatness and beauty of the country have been mentioned in the novel. *This Mournable Body is* as follows:

> It is my pleasure to introduce you to this fabulously beautiful country, our own Zimbabwe, a world of wonders for you to sample and of course enjoy. For you who are returning, hello again. Welcome! Mauya! Sibuyile!" you repeat in three official languages, for Tracey is concerned not to marginalize anyone and emphasizes that the Green Jacaranda greeting must align with the national language policy. (Dangarembga 2018, p. 169)

Finally, beyond the natural environment, the author confirms that Zimbabwe has a hidden treasure with a fascinating history that reveals its social, political, cultural, economic, and historical background. The speaker in the following extract, for example, states that there are many things that Zimbabwe does not show or tell the world. She asks, "Do you know, five of the ten best stone sculptors in the world are Zimbabwean?" (Dangarembga 2018, p. 180). This implies that Zimbabwe is still full of mysteries that humans cannot trace.

*2.3. Women's Concern of Nature*

Women are mostly vulnerable to the impact of environmental deterioration. As a result, they play a crucial role in managing natural resources at the home and community levels. The Green Belt Movement is a bold example of this fact in an African context.

In line with this, *This Mournable Body* reveals the concern of women about the degradation of the natural environment on the one hand and the plentiful natural resources seen everywhere, on the other hand. Tambudzai explains the two facets of Africa. The place that resembles a miserable ghost and the soil that is full of glittering minerals. She says, "On one side miserable ghosts, which are in fact maize plants, poke up from the earth" (Dangarembga 2018, p. 230). Tambudzai adds that "In front and behind you the soil glitters like pop stars' bling with mica, silicon, and crystals". This reveals the secret that Mother Earth had hidden within her womb. It also shows how Mother Africa is endowed with a plethora of natural resources.

Tsitsi Dangarembga also portrays the eroded land and devastated natural environment. She describes the mountains as: "The nearby mountains have, in the years since you last visited, grown as bald as underfed grandfathers" (Dangarembga 2018, p. 230). This confirms that the mountains have been eroded and defrosted over the course of time. As a result, Tambudzai declared that the environment was being destroyed. Though there were no forests, Tambudzai said that the landscape was so beautiful, and she described it as follows:

> Further away the grey granite of the Nyanga range lowers like a ridge of frowning eyebrows. You catch your breath as you greet these sentinels to your past, suppressing every twinge of regret at the events that brought you here or at the deed you are doing. (Dangarembga 2018, p. 230)

Environmental pollution and poverty are issues that Tsitsi Dangarembga depicts through her main character, Tambudzai. In the novel, the environment is heavily polluted and contaminated with a variety of filthy substances. She depicts poverty and environmental pollution in great detail. One thing, she indicates, is the dirty place where poor children live and spend their time. This place was very dirty and contaminated. She describes it as follows:

> The first sinks down and stays on the crumbling pavement. The sleeve of his jacket is a frayed rag. It flutters in the gutter. Beneath the cloth little dams of used condoms and cigarette butts build thick puddles of charcoal-coloured water. (Dangarembga 2018, p. 14)

She recounts about the area she was passing by, but in short, pollution and poverty are described boldly. She stated that the water was as dark as charcoal, which showed extreme pollution and contamination. Besides, there were condoms and cigarette butts around that displayed the contaminated environment. Tsitsi Dangarembga has also portrayed the griminess of a place in the city in another picture. The population, the poverty, and the filthiness of the city are described by her characters, as stated below:

> With the increase in travellers, in the terminus, and the roads around it are developing into markets. Women and adolescent boys sell airtime, vegetables, mazhanje and matohwe fruit, which are not stocked in supermarkets, and cheap Chinese biscuits, almost to the Green Jacaranda block doorstep. The city council has abandoned cleaning in favour of other pursuits. A pall of decaying leaves, pods, plastic wrappers, and peels is heaped at every corner. (Dangarembga 2018, p. 184)

Tambudzai also depicts the village and the toxic components that affect it. She demonstrates how pollution damages the village's natural environment and how it becomes contaminated. " . . . other youngsters bash at each other with cooking oil tins and pesticide pails and grind their elbows into each other's soft tissue. Past this fray your Mazda rumbles" (Dangarembga 2018, p. 229).

The issue of drought is another subject in the novel. People in rural areas are always hit by drought, but they have the ability to coexist with their natural environment. Tambudzai explains that though they endure the suffering of drought and famine, life is not easy for them. They really suffer; they are the only ones who experience true, genuine agony. However, they never become desperate; they believe that nature and God will help them. In contrast, people who live in cities do not know any real suffering. "That's why I always say please, please, someone from the rural areas. A person from those barren places without any rain. Those people know when God has given them something good. Because those people really know suffering" (Dangarembga 2018, p. 32).

Poaching and killing of animals is another issue in the novel. Africa is home to a wide range of animals. There are numerous wild animals and birds all over Africa, from east to west and north to south. All kinds, from the tiniest insects to the biggest mammals, live in Africa, including many endemic animals and birds that are only in Africa. These all make Africa the most wonderful continent in the world. Poaching animals, on the other hand, is the most common practice at the moment, and it is causing their extinction on a daily basis. People from Africa and some from all around the world come and hunt animals, commonly for business and for food. Tambudzai recounts this as follows:

> "The world's finest organic game," the tagline reads. "Eat only what you dare to pick, kill, or catch . . . The ultimate eco—in African." You know there is no such language as African, but you kept your expression constant as you read and continue to do so now. (Dangarembga 2018, p. 187)

### 2.4. Africa Natural Resources as "Milk from the Udder"

The systematic exploitation of the African natural environment is a continuing problem in the 21st century in many ways. As Betty and Wirth (1997, p. 300) clarify, the environment throughout the world is deteriorating, mostly because of human activity. Furthermore, the alleged progress has sped this up. Development, colonization, industrialization, and urbanization have all had a significant negative impact on the environment. Natural resources are typically exploited through unethical agricultural methods. Therefore, Tracy's participation in the ecotourism industry can be considered as one of the systematic exploitations of the natural environment, though the ecotourism industry has both positive and negative effects on the environment. This means, as Huggan and Tiffin (2015) illustrate, "Tourism can be beneficial if it is moulded to local needs and interests".

However, as Huggan and Tiffin (2015, p. 66) also suggest, "unsurprisingly, the sustainability of tourism is no less controversial than the sustainability of development . . . ". Because "tourist industry is neo-colonial extension of economic forms . . . " (Britton 1980, p. 149). Tracy's participation in the ecotourism venture demonstrates the tendency of promoting the neo-colonial economic form because her venture does not consider the interest of the indigenous society. On top of this, since Tracey's business was a village-based investment, it is difficult for Tracy to convince the local people. So, what Tracy did is tell Tambudzai to lead the project and facilitate it. In order to control the village, Tambudzai was chosen to convince the people and lead the job. That is the reason Tracy states, "You have a rural background, Tambu. You embody it. That's how you can, if you're up to it, take on the brand we created on the farm. This time in a village . . . Pedzi snorts, 'Queen of the village!'" (Dangarembga 2018, p. 225).

As a result, Tambudzai tries her best to convince the people. However, they still do not accept her wholeheartedly because they suspect any person who has a connection with white people. The image of whites in rural areas was totally negative. That is why Tambudzai's mother expresses her emotions and tells her that the whites are dangerous and can not be trusted. She asks her daughter to be careful and free herself from white people and their jobs. She warns Tambudzai to be independent by saying the following:

> "Who is this Stevenson?" she inquires, enjoying the shock she causes you with her alertness. "Do we know that family? Are they one of our white people who farm in our parts of the country?"

You hesitate.

Mai stands still. (Dangarembga 2018, p. 237)

The novel also addresses the impact of neo-colonialism on women, showing its indirect impact on Africans in general and women in particular. On the other hand, it shows how women resist the unjustified exploitation committed by white people. Tambudzai's mother has stated how it all happened; she expresses her threat to the whites, and she even believes that they are problems because they cannot be trusted. In one way or the other, they work to benefit themselves, not the local people. They used you to accomplish what they planned. They treat you until they get what they want from your land and your environment, then they leave you alone. They sabotage the natives by taking their wealth from the natural environment and leaving it regraded. Thus, since Tracy's dream was to expand her control of the village, Tambudzai's mother was unhappy. This shows her protest against the extension of the exploitation of the natural environment. In short, to make sure that Tambudzai is working with the right person, she warns her as follows:

> "White people are a problem," she remarks. "You can only work with them if you know them. That's why we prefer to do things with our own ones. You have to know this Stevenson properly to work with her, my daughter. Play cunningly. If her family is not from these parts, how can you know her? And her, what does she want with you if she doesn't know you? (Dangarembga 2018, p. 237)

Furthermore, the novel also explores women's socioeconomic vulnerability in post-colonial Zimbabwe. Tambudzai's mother believes that Tambudzai's approach to the whites is the reason. Tambudzai is a college-educated young woman who did not get a job. During her education, she spent most of her time with white students. However, after her graduation, she started to live in a city without a job. As a result, her mother did not want her daughter to be around white people. Tambudzai expresses her mother's feelings: " . . . sadness about white people again, Tambudzai. Isn't that why you have been nothing all this time, because of too much of those people? Leave them alone. Go and find your own thing. That is what I can tell you." (Dangarembga 2018, p. 247).

Mai, as stated earlier, seems suspicious to the whites. She does not have a positive image at all. As a result, she frequently makes comments about white people. The villagers' view, especially her mother's attitude to the whites, is kind of distrustful, though Tambudzai calls it just boasting and tries to express her positive outlook on the whites and on Tracey in particular.

> "And we will be paid? Each for doing what we do? All of us, properly?" your mother asks warily. "These white people, they say something and they do it too, but the way they do it, you just never know what it is they first said they were doing." . . . "I tell you, Mai, I now know them," you say. After so much tension you are unable to resist a little boasting . . . "Do you forget I spent all those years at the Young Ladies' College of the Sacred Heart? I know our white people. And I have worked with her for so long, I can say I now know my boss better and I also know what she is talking about." (Dangarembga 2018, p. 237)

The author shows that Mia seems very likely to suspect the Whites. The reason is that Tracy, after controlling vast places, also dreams of expanding her territory and controlling many rural land areas, as her father does. To attract clients and advertise her investment, she talks about the opening of new sites and what the name could be, and she named it "Opening Eco Special".

While they open new locations, they have partners from all over the world. "The Amsterdam partner says it's fine in principle, but they're asking for a discount" (Dangarembga 2018, p. 257). This means that they have intertwined business activities that take advantage of Africa. This indicates there is still exploitation of Europeans in Africa. The way they control the land may be different, such as paying bribes, misleading the leaders, and creating sabotage that benefits them but not the Africans. On the other hand, they elucidate that the village and the women are of great value to their investment.

Tambudzai clarifies this by saying that "You were right, Tracey" (Dangarembga 2018, p. 257). On the other hand, Tracey magnifies Africa as a bloody continent.

> "Africa", amplifies Tracey. "How're we going to add value to a bloody continent? Oh, why did they have to go onto those farms? But let's not go into it again. It's just that they had value on the farm. They expect the village to top it.". . . . Well, for Amsterdam, obviously the farm is the farm. The village, well that means, for them, something different, maybe not as interesting." . . . "Different?" you repeat. "Interesting. That's what we're working to make it." (Dangarembga 2018, p. 257)

The exploitation and destruction of indigenous trees is another issue in the novel. These indigenous trees have many benefits. However, for Tracy, the value of these indigenous trees is in making furniture. She does not see its value to the ecosystem. She encourages the people around her to materialize these trees. She explains that the furniture is local, and she is proud of having all these local materials. "Everything's one hundred percent local. The proper boardroom fixtures will be too, when we get them" (Dangarembga 2018, p. 185). This indicates that Tracy does not care about the destruction of the environment, but she pretends to beconcerned about it. However, in reality, she does not pay attention, for example, about the trees as long as they are used for her personal purposes. Therefore, in order to own local materials, she encourages people to cut down trees, which is the opposite of what she said about climate change.

Another subject that is mentioned in the novel is the introduction of the Chinese. Neo-colonialism is still prevalent in many African countries in many forms as stated earlier. Aside from these Westerners, China, as a country, has a strong interest in occupying Africa. Consequently, the Chinese government has shown an interest in having a strong relationship with many African countries. As a result, the Chinese participate in different activities. Thus, Africans have their own perceptions of what and how the Chinese government operates. The author makes numerous references to China in the novel, such as "cheap Chinese biscuits" sold on the streets of Harare, Zimbabwe, to show how the Chinese control the market. Besides, the narrator illustrates the introduction and systematic exploitation of the Chinese in Africa as follows.

> The government's working on the Chinese, which promises to be a great market, but, it's all bilateral with the Asians, you know. Looking east. Surely, you've heard and can tell me the implications of that?" You nod noncommittally, trying to recall the phrases in the newspapers you browsed through occasionally at your cousin's. "The Chinese are interested in governments, not people", continues Tracey. "That being the case, we can't get to them, especially given our funding sources. So, in principle it's your Europeans. That's an established market so, for the time being, we can't change our continent's 'single singular thought'. Nature, which naturally also means weather. Sun. As time goes on and things get worse or better, we'll strategize for the next phase." (Dangarembga 2018, p. 187).

China's interest in Africa may have various objectives, as stated earlier. According to Tracey, "the Chinese are interested in governments, not people" (Dangarembga 2018, p. 187). Though she is a white European woman, she depicts the truth that China is exploiting Africa in many ways. However, the intention of the writer here is to highlight the competition between China and the Europeans. Tracey claims that "That being the case, we can't get to them, especially given our funding sources. So, in principle it's your Europeans. That's an established market so, for the time being, we can't change our continent's 'single singular thought'." (Dangarembga 2018, p. 187).

Controlling the natives' land is another subject. Neocolonialism has caused poverty and left the natives without land in their country. As a result, in post-independent Africa, governments still give land to the whites in the name of investment. This is reflected in the novel *This Mournable Body*, which states, "Since the government started giving people land in places, we thought they were only for Europeans" . . . "It was my aunt's place", you say. "She was given it by her employer. He went to Australia" (Dangarembga 2018, p. 12).

There are conflicts between natives and white settlers in post-colonial Zimbabwe, which shows the legacy of colonialism. The reason is that many white people own major

land areas, and they use these places for different reasons. They use them for farms, ranches, and so on. As in this novel, villages were also tourist sites, such as those owned by Green Jacaranda Safaris (Tourist Site Villages). However, the natives have reclaimed these villages from the whites. As a result, conflicts escalate when such situations occur. White people consider themselves to be the owners of the land, and they complain that indigenous people are invading it:

> The boss raises her glass to her lips, does not drink, reaches for a carton, and mixes the wine with orange juice. "Well, there's been trouble. Some of those . . . thugs . . . skellems who call themselves ex-combatants, or war vets . . . they've occupied the rondavels. They're hunting the game. And they're camping in our tourists' village! It's not like we're not the only ones. There's a whole lot of these . . . these invasions. I've been thinking how we can go on. Just the other day I realized, we're safest in a real village. If we can get one". Your boss grunts despondently. "That was their philosophy during the war too, wasn't it? Being part of the village. For a safety strategy." (Dangarembga 2018, p. 224)

The conflict between the white settlers and the local politicians has different facets. In the next extract, Tracey and Tambudzai speak about the situation of the reclamation of the ex-combatants of the land which was taken by the white settlers. Tracey asserted angrily that everything done was a racist and unacceptable publication. She belittles the people's protests and calls it a "bloody war." She was furious that the indigenous people were reclaiming their land. Tracey speaks about the situation as follows:

> . . . "In principle, it's a racist publication. You can't dignify it by calling it a newspaper." . . . "It's absolutely unbelievable," . . . "It's like the . . . the . . . bloody war," your boss says, turning the paper over so that the sports page featuring two top cricketers is visible. "They're singing. They're triumphant? They've invaded lots more places. Because the Old Fossil ordered it. It's been part of his plan all along. People used to say that at the agency, but I stood up for this country. I couldn't believe it. Can you believe they were ordered to do it, to go out and destroy honest, hardworking people's homes? (Dangarembga 2018, p. 253)

As a result, while Tracy was accumulating wealth by investing in the ecotourism industry, many people still live in harsh poverty in the villages she wanted to invest in. This happened because of unsustainable development and the ecotourism industry. Many stayed with their families, who were considered educated and from the Western world. To express how impoverished they were, Tambudzai's mother shares her expectations of her children and her comments by saying, "We nearly died of hunger while we waited for that to happen" (Dangarembga 2018, p. 128).

She stated that, because the whole family was living in poverty, Tambudzai's mother did not want to accept that her daughter was unemployed. It was devastating for her to learn that her daughter was unemployed for a period of time. The mother criticizes her daughter, and she expresses her emotions about what she feels and what society thinks about her.

> What we heard all the time is that you were not working. That's what was said, that that degree of yours was just a piece of paper sitting, silently rotting. And I just kept on thinking, that's the paper. What about that daughter of mine? Tambudzai, even when Lucia sent me worse messages about you, I just kept it in my mind, surely my daughter is not sitting there like paper that has been written on and finished. I said my daughter can't be sitting there just like that, rotting. (Dangarembga 2018, pp. 245–47)

At last, Tambudzai's mother, Mai, is furious about her children's lives. She talks about the failures of her children repeatedly. The one who does not achieve her dream and starts working; the one who is in Ireland and he does not send money, the last one who was in a fight and lost her leg. This is a common crisis in many post-colonial societies. There

is fragmentation of families because of poverty, war, and conflict. Thus, the author has presented all these issues in many ways in general.

### 3. Conclusions

Based on the analysis made, the novel, *This Mournable Body,* has many ecofeminist issues. Firstly, it extensively depicts the interaction between women and the natural environment. The novel depicts women as gardeners and gardening experts. Furthermore, women in rural areas are portrayed as mostly engaged in agricultural labor. They devote their lives to agricultural activities and sustaining their families. While they do this throughout their lives, they are very concerned about the fertility of the land. They value natural products and want to preserve and safeguard nature in its natural state. As a result, while their lives are dependent on nature, they also protect and preserve nature in its natural state. In this sense, the novel demonstrates the interdependency between women and nature in Africa.

Moreover, the novel also depicts how women from various backgrounds perceive nature. In this case, the way Tracy, a white business woman, Tambudzai, a Zimbabwean college graduate, and the other women originally from Zimbabwe relate themselves to the natural environment portrayed in the novel is evident in many ways. For Tracy, for example, the natural environment is a source of income only. She believes that nature's value is to generate money. For this reason, she participates in the eco-tourism industry, which enables her to monopolize vast areas as tourist sites and employee many local workers, such as Tambudzai, participating in this industry may have a positive side since ecotourism as a business has a lower impact on climate crises. However, since Tracy's dream is to accumulate wealth, she does not care about the destruction of the natural environment. She does not even know the mesh ecosystem. As long as there is good sun and conducive weather, that is enough for her. She believes that that is the only essence of nature. By doing this, Tracy intends to expand her tourist sites to other African regions. She even has partners from Europe. While she does this, her local employees support her as tourist guides and facilitators. Thus, the author is addressing Tracy's negligence of the natural environment.

On the other hand, Tambudzai perceives nature in different ways. Though it was not her intention to be a tour guide in her country, she traveled with assistance from a travel agency and visited many places, reflecting on the natural environment from a variety of points of view. For this reason, Tambudzai believes that nature is a source of happiness, freshness, and recreation. Moreover, she conveys that nature is the source of life-reviving power. Furthermore, she believes that she is part of the natural elements. She considered herself a member of the community of the earth. She also depicts the beauty of the biodiversity in Zimbabwe. Finally, Tambudzai expresses her concern about the degradation of the natural environment in many ways.

Similarly, the indigenous female farmers consider nature as a means of their subsistence. Hence, they cultivate gardens and are involved in agricultural activities. By cultivating gardens that grow all types of fruits, vegetables, and some other important products, they are very popular throughout the country. This shows their ability to sustain life in harmony with the natural environment. On top of that, it shows that these women who grow gardens everywhere play a pivotal role in reducing the climate crisis. They do not destroy and degrade the natural environment. It shows their contribution to balancing the ecosystem. Women in the rural areas are also very curious about the healthy environment. They avoid using fertilizers and other chemicals in order to save their land from any pollutants. As a result, they produce organic foods, and they are happy with it. Therefore, in this way, women are represented as protectors of the earth and the natural environment in general.

Likewise, the indigenous societies generally believe that nature is God's home. They protect nature because they are spiritually connected to it. Accordingly, they do not want to cause any harm to the ecosystem. They, as human beings, are part of the fauna and flora

of the earth. They live in the natural environment as part of the other inhabitants of the earth. Thus, showing that nature is their home.

The novel also discusses how neocolonialism and capitalism continue to exploit Africans, particularly women, and their natural resources. The locals, especially women, are still poor and labor for the whites. The whites systematically exploited the natives and their lands, and this makes the local women resentful and suspicious of whites. In a nutshell, Tracy's and other white businessmen's investments, as well as the entry of the Chinese into many sectors, have been portrayed in the novel. This indicates capitalism and neocolonialism remain ingrained in African society.

While this is a fact, the novel demonstrates the burden of women in many ways. Tsitsi Dangarembga proves that society in post-colonial Zimbabwe is contradicted by what is happening in the nation. People who fought for freedom have been left without further help. Those who are educated are still unemployed. All of this burden falls on women's shoulders. It shows the confusion and contradictions in Zimbabwe's present-day society.

In general, Tsitsi Dangarembga portrayed those women, having a variety of backgrounds, as having different ways of understanding nature. In addition, she has revealed the superiority of African women in gardening, farming, and preserving nature and the land. Furthermore, she demonstrates African women's hard work and endeavors to sustain life and grow in their own way. Moreover, she shows the way Africans understand the value of their natural environment in general. Finally, she reveals the exploitation of the African natural environment in the 21st century in the name of investment and trade, while poverty has rooted itself in the lives of indigenous societies.

**Author Contributions:** Conceptualization: N.M.G., A.D.A.; methodology: N.M.G., A.D.A.; formal analysis: N.M.G., A.D.A.; writing—original draft preparation: N.M.G.; writing—review and editing: N.M.G., A.D.A.; supervision: A.D.A. All authors have read and agreed to the published version of the manuscript.

**Funding:** This research received no external funding.

**Conflicts of Interest:** The authors declare no conflict of interest.

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
