# Peer review of "Women’s Perceptions of Nature: An Ecofeminist Analysis of Tsitsi Dangarembga’s This Mournable Body"

_humanities, doi:10.3390/h11060159_

Round 1

Reviewer 1 Report

The paper needs to be re-read with a focus on the way in which the phrases and the arguments are used. In some parts despite the use of conclusive phrases, none of the arguments invoked logically results from the previous phrases. For example, someone can only look at the Introduction and see the paragraph three. Starting with 'Thus' and using 'As a result' in its body this paragraph does not demonstrate anything and the phrases are not logically connected. Furthermore, why using the ecofeminist perspective and not another one? What is the reason behind the methodology adopted?

- A lot of quoting material is used without a critical engagement with it.

- What is the issue of 'nature women'?

Another example from the introduction: the six paragraph - how is the question of 'other subordinate groups' different from that of the 'indigenous people'? Some clarifications are required. 

Section 2. Results and Discussions: for example, what is the Purple Hibiscus and who is Adichie.

This type of omissions, non-logically introductions of phrases and references are repeating in the body of this paper.

I am afraid that the reader cannot fully follow and read this paper in this current state.

Reviewer 2 Report

This paper currently reads more like a list of ecofeminist issues in This Mournable Body than an analysis that can point to larger critical issues around climate change, the allocation of resources, and the importance of combining feminist perspectives with eco-criticism. I think a strong introduction addressing the overall importance of this analysis would be helpful and would ground the analysis. 

I'm unclear about the disciplinary home of this analysis as at times it sounds more like an social science paper than a humanities paper. 

Reviewer 3 Report

I think the idea os a good one, but I believe the author needs to expand their knowledge of ecofeminism, first. Most of the references are older and grounded in colonial authors work. More recent ecofeminism work, and the author cites a few, focus upon non-colonial writers/theories and critiques. There is a who collection grounded in Africa-centric ecofeminism that could have brought some very interesting insights to the analysis (and I write this as a settler individual myself). I think as well that ecofeminism has a stream on literary critique that might have been drawn upon.

I also found the organization of the main analysis problematic. A section is quoted and a sentence indicates this reflects XXX in ecofeminism. There is no real analysis. Nor is there any organization in the themes presented. Many could be interconnected or have other with no connection or linkage. This was not enlightening on the themes in the book.

I think this manuscript has potential but it needs more thoughtful grounding in a more sophisticated understanding of ecofeminism, better analysis and a more thoughtful discussion.

Round 2

Reviewer 1 Report

I have no further comments on this paper.

Reviewer 3 Report

Well done on the revisions. This is a much stronger

article.